# Significant Improvement in Bioavailability and Therapeutic Efficacy of Mebendazole Oral Nano-Systems Assessed in a Murine Model with Extreme Phenotypes of Susceptibility to *Trichinella spiralis*

**DOI:** 10.3390/pharmaceutics17081069

**Published:** 2025-08-19

**Authors:** Ana V. Codina, Paula Indelman, Lucila I. Hinrichsen, María C. Lamas

**Affiliations:** 1Instituto de Genética Experimental, Facultad de Ciencias Médicas, Universidad Nacional de Rosario, Santa Fe 3100, Rosario S2000KTR, Argentina; 2Carrera del Investigador Científico de la Universidad Nacional de Rosario (CIC-UNR), Universidad Nacional de Rosario, Maipú 1065, Rosario S2000CGK, Argentina; 3Área Parasitología, Facultad de Ciencias Bioquímicas y Farmacéuticas, Universidad Nacional de Rosario, Suipacha 531, Rosario S2002LRK, Argentina; 4Instituto de Química Rosario-Consejo Nacional de Investigaciones Científicas y Técnicas (IQUIR-CONICET), Suipacha 570, Rosario S2002LRK, Argentina; 5Departamento de Farmacia, Facultad de Ciencias Bioquímicas y Farmacéuticas, Universidad Nacional de Rosario, Suipacha 570, Rosario S2002LRK, Argentina

**Keywords:** mebendazole, nano formulations, animal model, oral pharmacokinetics, trichinellosis, anthelmintic efficacy

## Abstract

This study aimed to analyze whether the enhancement of the biopharmaceutical efficiency of mebendazole, a poorly water-soluble anthelmintic drug, significantly improves its antiparasitic activity in a murine model of trichinellosis. **Objectives**: Two advanced oral formulations were developed, polyvinyl alcohol-derived nanoparticles (NP) and β-cyclodextrin citrate inclusion complexes (Comp), both employing mebendazole as an anthelmintic agent. The primary objective of this work is to treat trichinellosis, an infection with severe chronic effects. **Methods**: The physicochemical characteristics as well as the in vivo performance of the NP and Comp formulations were assessed. The in vivo studies involved the bioavailability analysis, comparing drug absorption between the pure drug and the novel formulations, as well as the in vitro anthelmintic activity and in vivo therapeutic efficacy against *Trichinella spiralis* encysted muscle larvae. The in vivo efficacy was evaluated during the parenteral stage of *T. spiralis* infection in male and female mice from two genetically distinct lines differing in mebendazole pharmacokinetic parameters and susceptibility to the parasite. **Results**: The formulations exhibited smaller particle sizes and improved dissolution properties compared to pure MBZ. The pharmacokinetics studies indicate that NP and Comp significantly improved MBZ bioavailability. Both NP and Comp significantly increased mebendazole’s anthelmintic activity against the encysted parasites, which would be attributed to the improved MBZ absorption. The formulations overcome the drug’s poor solubility and low bioavailability limitations, resulting in a higher plasma concentration of the active drug, even at low doses. **Conclusions**: These findings suggest that the newly designed mebendazole formulations are suitable for treating *T. spiralis* chronic infection and highlight a potential improvement in the pharmacological treatment of trichinellosis.

## 1. Introduction

Trichinellosis is a parasitic zoonosis caused by *Trichinella* spp., including *Trichinella spiralis*, which humans contract primarily through consuming undercooked or raw meat contaminated with encysted larvae [1]. It is a significant concern for public health, especially in regions with endemic outbreaks such as Latin America [2,3]. It is worth noting that trichinellosis is a reemerging disease and the seventh most important global parasitic infection [4,5]. The disease is characterized by its chronic nature, posing a significant challenge to healthcare systems worldwide. Diagnosis and treatment in the early stages are challenging, and effective procedures to limit infection have not been found to date. Miguez et al. [6] reported an interesting study on human cases across 38 countries between 2005 and 2022. The results underscore the importance of the proper management of this type of disease, given consumers’ sensitivity to potential infections from the products they consume.

The pharmaceutical therapy available for treating these parasitic diseases consists of several anthelmintic compounds, including mebendazole (MBZ) [7]. MBZ, a broad-spectrum anthelmintic compound used for treating and controlling several parasitic infections, is currently on the World Health Organization Model List of Essential Medicines. It is a member of the benzimidazole family of drugs and exhibits broad-spectrum anthelmintic activity. The WHO Model List of Essential Medicines (21st List—2019) already includes albendazole, MBZ, and praziquantel as intestinal anthelminthics. MBZ is commercially available in various pharmaceutical dosage forms to meet the diverse needs of patients. The most common formulations include chewable tablets (typically containing 100 mg of the active ingredient), conventional oral tablets, and oral suspensions (usually at a concentration of 20 mg/mL or 100 mg/5 mL). In some regions, such as Europe and Latin America, dispersible tablets are also available, particularly intended for pediatric use. These dosage forms have received regulatory approval from agencies such as the U.S. Food and Drug Administration (FDA) and the European Medicines Agency (EMA), following the evaluation of their safety, efficacy, and quality [8,9,10]. The anthelmintic activity of benzimidazole 2-carbamates has been related to their selective antimitotic activity due to the preferential binding of these agents to helminthic tubulin over mammalian tubulin [11].

Mebendazole exhibits polymorphism, with each polymorphic form showing a significantly different aqueous solubility: 9.8 µg/mL for polymorph A, 71.3 µg/mL for polymorph B, and 35.4 µg/mL for polymorph C [12]. According to the United States Pharmacopeia (USP), all MBZ forms are considered practically insoluble in water. This compound is classified as a Class II drug in the Biopharmaceutical Classification System (BCS), characterized by its high permeability but low solubility in aqueous media [12]. Its unique chemical structure and mechanism of action make it highly effective against various parasitic infections, including nematode infestations such as trichinellosis. It is usually well-tolerated but may present common side effects such as stomach pain, diarrhea, nausea, and dizziness. Its oral bioavailability is approximately 20%. MBZ’s poor oral absorption is primarily due to its extremely low aqueous solubility, despite having high intestinal permeability. In this context, designing and developing new pharmaceutical formulations is decisive in achieving optimal results after oral administration.

Our laboratory has focused on developing new β-cyclodextrin (β-CD) derivatives. Cyclodextrins (CDs) are cyclic oligosaccharides composed of six, seven, or eight D-glucopyranose units, corresponding to α-, β-, and γ-CDs, respectively. Owing to the chair conformation of these glucose units, CD molecules adopt a toroidal, truncated, cone-like structure, characterized by a hydrophobic inner cavity and a hydrophilic outer surface [13]. Cyclodextrins (CDs) are well known for forming inclusion complexes with various drugs, enhancing their aqueous solubility. Consequently, natural and modified CDs have been widely studied to improve the solubility and bioavailability of hydrophobic drugs, reducing the need for organic solvents or surfactants. Chemical modification of CDs, particularly the random substitution of hydroxyl groups, significantly increases solubility and enhances drug delivery efficiency [14]. Cyclodextrins contain 18 (α), 21 (β), or 24 (γ) hydroxyl groups, which are chemically modifiable. Among them, C6-OH groups are the most reactive, while C3-OH groups are the least. However, reactivity differences are modest and highly dependent on reaction conditions, making the selective derivatization of CDs challenging [15,16,17]. The new β-cyclodextrin derivative (citrate-β-cyclodextrin) generated in our lab exhibits enhanced water solubility, making it an exceptional carrier for incorporating hydrophobic compounds [18].

Nanoparticles (NPs) are materials with dimensions in the nanometer (nm) range. Their small size endows them with unique physical, chemical, and biological properties, making them valuable in medical, pharmaceutical, and environmental applications. The increased surface area of nanoparticles enhances their solubility and absorption, further improving the bioavailability of active compounds. Nanoparticle-based formulations utilizing hydrophilic polymers, such as polyvinyl alcohol (PVA), serve as excellent delivery systems for orally administered active compounds [19].

Thus, the primary goal of this study was to develop nanoparticles and inclusion complexes of MBZ to enhance its solubility, dissolution profile, and bioavailability, thereby improving its therapeutic efficacy as an anthelmintic [20]. The in vivo activity was evaluated during the parenteral stage of *T. spiralis* infection in males and females from two mouse lines differing in the novel formulations’ pharmacokinetic parameters and susceptibility to the parasite. Using phenotypically defined animal models is a significant step forward in understanding the role of the host genetic background in resistance to parasitism and response to treatments. This approach may help identify the genes involved in the trait, as genetic factors can influence a drug’s efficacy and the likelihood of an adverse reaction. Comparative analysis of extreme response phenotypes, such as those used in these experiments, is instrumental in studies of the host–parasite relationship [20,21,22,23] and has been proposed as the procedure of choice in pharmacogenomics due to the phenotypes’ “Mendelian” nature [24]. Furthermore, it plays an important role in evaluating the efficacy and safety of new pharmaceutical devices as therapeutic alternatives for medical and veterinary use, potentially leading to more effective treatments.

## 2. Materials and Methods

### 2.1. Materials

Mebendazole (MBZ) and β-cyclodextrin (β-CD) were supplied by Sigma-Aldrich Chemie GmbH (Steinheim, Germany). RPMI 1640 was obtained from Sigma-Aldrich Chemie GmbH (Steinheim, Germany), whereas fetal bovine serum and gentamicin were purchased from Klonal (Buenos Aires, Argentina). All other chemicals were of analytical grade.

### 2.2. Nanoparticle Synthesis

MBZ (50 mg) was dissolved in formic acid (2.4 mL) and sonicated until complete dissolution was reached. The resulting solution was dropped over an aqueous solution of polyvinyl alcohol (PVA, 10 mL, 0.5% *w*/*v*) under magnetic stirring at 1000 rpm for 10 min. Then, the suspension was dried using a Mini Spray Dryer Buchi B-290 under the following controlled conditions: inlet temperature, 130 °C; outlet temperature, 60 °C; airflow, 30 m^3^/h; feed rate, 5 mL/min; and an aspirator set to 100% [12].

### 2.3. β-Cyclodextrin Citrate and Inclusion Complexes’ Synthesis

The β-cyclodextrin citrate (C-β-CD) and inclusion complexes were synthesized following the procedure outlined by García et al. [25]. Briefly, citric acid (10.57 mmol, 2.03 g) was dissolved in water (1.2 mL), and β-CD (1.76 mmol, 2.00 g) was added in a 1:6 molar ratio. The reaction mixture was refluxed at 100 °C for 6 h. Then, the product (C-β-CD) was precipitated, employing isopropanol, and centrifuged at 6000 rpm. The precipitate was washed twice with isopropanol to remove unreacted citric acid. Finally, C-β-CD was dried for 24 h at 60 °C, mortared, and stored in a tightly closed bottle until use. The inclusion complexes (Comp) were prepared using the spray drying method. MBZ (0.165 g–0.56 mmol) was dissolved in formic acid (10 mL) then C-β-CD (0.717 g–0.56 mmol) and water (20 mL) were added to the solution. The resulting solution was dried in a Mini Spray Dryer Buchi B-290. The drying conditions were as follows: inlet temperature, 130 °C; outlet temperature, 70 °C; airflow, 38 m^3^/h; pump efficiency, 12% (flow rate ~5 mL/min); vacuum cleaner, 100% [26].

Since both formulations were developed for oral administration, hazardous solvents should be avoided, and only Class 3 solvents, such as formic acid, should be used, which are solvents with a low toxic potential according to ICH guideline Q3C by the European Medicines Agency. The objective of this guideline is to recommend acceptable amounts for residual solvents in pharmaceuticals to ensure patient safety.

The absence of formic acid in the final formulations was confirmed by ^1^H NMR spectroscopy, based on the non-detection of its characteristic signal at 8.2–8.4 ppm in the ^1^H NMR spectrum. This technique offers a precise and direct method of analysis that does not require prior derivatization. The corresponding spectra have been included in Appendix A (Figure A4 and Figure A5) [27,28].

### 2.4. Particle Size Determination

The particle size of the NP and inclusion complexes was determined by imaging using scanning electron microscopy (Leitz AMR 1600 T [Amray, Bedford, MA, USA]), with an acceleration potential of 20 kV. Samples were sputter-coated with a gold layer to make them conductive. Then, to determine the particle size, scanning electron microscopy images were analyzed using the ImageJ 1x software, a Java-based graphic design program dedicated to image analysis, to estimate size. The particle size of nano-formulations was determined by using a Nano Particle Analyzer Horiba SZ-100 (Kyoto, Japan). Samples were diluted (1:30) in filtered distilled water before measuring.

### 2.5. Solubility and Dissolution Studies

The solubility of MBZ in each formulation was assessed by adding an excess of the drug to sealed vials containing 5 mL of 0.1 N HCl, followed by agitation at 180 rpm for 72 h in an orbital shaker. Samples were then filtered through 0.45 μm PVDF filters, and drug concentrations were quantified by UV spectroscopy at 285.0 nm using a Shimadzu spectrophotometer, Kyoto, Japan. Dissolution studies were conducted using a Hanson Research SR8 8-flask bath with USP apparatus II, Chatsworth, California, Estados Unidos (paddle) at 75 rpm, in 900 mL of 0.1 N HCl at 37.0 ± 0.1 °C.

Dissolution efficiency (DE) was calculated according to the method described by Khan [29], as the percentage ratio of the area under the dissolution curve up to time *t* to the area of the rectangle representing 100% dissolution over the same time (Q_100_), using the following equation:DE = (AUC × t_0_/Q_100_ × t) × 100

The area under the dissolution curve was calculated using GraphPad Prism 9.5 (GraphPad Software, San Diego, CA, USA).

Solubility and dissolution experiments were carried out in triplicate. Following USP procedures, the dissolution medium for MBZ was 0.1 N HCl, which contains 1% sodium lauryl sulfate, an anionic surfactant [30].

### 2.6. Animal Model

Adult CBi/L and CBi+ mice (80–90 days old) from the Animal Facilities of the Instituto de Genética Experimental, Facultad de Ciencias Médicas, Universidad Nacional de Rosario (CBi-IGE stock), were used. These lines, derived from 160 generations of divergent selection for body conformation (theoretical inbreeding coefficient: 0.99), differ in body shape and weight: CBi/L (low body weight, long tail) and CBi+ (high body weight, long tail), with respective weights (mean ± SEM, g) of 29.9 ± 0.23 (males) and 27.2 ± 0.21 (females) for CBi/L, and 48.3 ± 0.48 (males) and 44.9 ± 0.55 (females) for CBi+. These lines also exhibit differential responses to infection with increasing doses of *T. spiralis* [31]. Previous experiments demonstrated that these lines also differ in response to infection with increasing doses of *T. spiralis* [32]. Significant differences in infection intensity were observed among genotypes. Although muscle larval burden increased with higher infective doses, the magnitude of this increase varied significantly between lines. Based on intestinal and muscle parasite loads—key indicators of host resistance—the CBi/L line was classified as resistant, while CBi+ was identified as the most susceptible genotype.

All the experiments with animals were performed during the first half of the light cycle. Mice had access to a complete balanced feed for rats and mice (Gepsa Feeds, Grupo Pilar S.A., Pilar, Buenos Aires, Argentina) and water ad libitum. At the end of the experiment, mice were euthanized by inhalation of 70% CO_2_ using a chamber prepared ad hoc or by cervical dislocation.

Mice were handled according to the institutional regulations (Facultad de Ciencias Médicas, Universidad Nacional de Rosario, permit number 1926/2020), which comply with the guidelines issued by the Institute for Laboratory Animal Resources, National Research Council (2011).

### 2.7. Parasite

*T. spiralis* was generously provided by Dr. Maria Dalla Fontana (Laboratorio de Zoonosis, Laboratorio Central de la Red Provincial de Laboratorios, Dirección de Bioquímica y Farmacia, Santa Fe, Argentina). This parasite has been maintained in CBi mice since 2006. The strain was genotyped as *T. spiralis* using multiplex polymerase chain reaction (Dr. Silvio Krivokapich, Departamento de Parasitología, Administración Nacional de Laboratorios e Institutos de Salud ‘Dr. Carlos Malbrán’, Buenos Aires, Argentina, personal communication).

The L1-infective larvae used in the infection were recovered by artificial digestion from the muscles of mice from line CBi, infected 3–4 months earlier for that purpose, as described previously [32].

### 2.8. In Vitro Analysis of the Anthelmintic Activity of the MBZ Formulations

#### 2.8.1. Preparation of *T. spiralis* Female Worms

CBi mice were orally infected with 10 L1 *T. spiralis* larvae per gram of body weight and sacrificed on day 6 post-infection (intestinal phase) as described previously [32]. After euthanasia, the small intestine was excised, segmented into equal lengths, and placed in Petri dishes containing 8–12 mL of incubation medium (0.85% NaCl with 250 μg/mL gentamicin). Each segment was opened longitudinally, gently agitated in the medium, and incubated for 4 h at 37 °C in 5% CO_2_. Following incubation, intestinal pieces were rinsed with medium and discarded. The medium containing released parasites was collected, centrifuged at a low speed, and the pellet was resuspended in 2–3 mL of RPMI 1640 supplemented with 250 μg/mL of gentamicin and 10% fetal bovine serum. Female worms were then isolated under sterile conditions for further assays.

#### 2.8.2. Preparation of the Antiparasitic Solutions

As described elsewhere [18], MBZ, NP, and Comp stock solutions were prepared at a 10 mg MBZ/mL concentration in dimethyl sulfoxide (DMSO). Briefly, the working solutions were formulated with RPMI 1640 medium containing gentamicin (250 μg/mL) and DMSO (up to a maximum of 2%) to obtain a final concentration of 500 μg/mL of MBZ. They were stirred at room temperature for 24 h and then filtered through sterile ethyl cellulose 0.2 μm pore filters (Minisart^®^, Sartorius Stedim Biotech, Göttingen, Germany).

#### 2.8.3. In Vitro Parasitic Assay

The entire procedure was conducted under sterile conditions as described by Priotti et al. [19]. Female *T. spiralis* worms obtained as above were incubated overnight in RPMI 1640 supplemented with 250 μg/mL of gentamicin and 10% fetal bovine serum at 37 °C in 5% CO_2_. For the assay, 10–12 females per well were placed in 24-well plates containing the tested MBZ formulations supplemented with 10% fetal bovine serum. The MBZ solution served as the positive control, and the culture medium with the solvent served as the negative control. Worms were incubated at 37 °C in a humidified 5% CO_2_ atmosphere for 30 h and observed under an inverted microscope at 2, 4, 7, 24, and 30 h. Viability was assessed at each time point by motility and morphology to classify worms as live or dead. The number and motility of newborn larvae were also recorded. Median worm survival was analyzed using survival curves. Experiments were performed in duplicate, with data corrected against the negative control.

### 2.9. Pharmacokinetic Analysis

A suspension of pure or formulated MBZ was prepared by mixing 50 mg of powder with 1 g of glycerin and sonicating to homogenize. Aliquots were transferred to Eppendorf^®^ tubes to achieve a dose of 15 mg MBZ/kg average body weight for each mouse line and sex. Separately, 2.6 g of gelatin powder was dissolved in 10 mL of hot water (60 °C), and 100 μL of this solution was added to each tube before solidification. The mixture was then sonicated in a 60 °C water bath, and 40 mg of powdered commercial chow was added to enhance palatability and ensure complete ingestion. This procedure was developed in our lab and was published in Codina et al. [33].

The final pharmaceutical dosage form contained either pure MBZ or the formulation, gelatin, and food powder. Glycerin and gelatin are GRAS (Generally Recognized as Safe) excipients listed as inactive ingredients by the U.S. Food and Drug Administration. Finally, the formulations were stored at 4 °C until administration.

CBi/L and CBi+ males and females were fasted for approximately 12 h, with free access to water (n = 93–99 per line and sex), and randomly divided into three groups to receive a single oral dose of pure MBZ, NP, or Comp prepared as described. Following the procedures described by Codina et al. [34], three to four animals per formulation were anesthetized after the drug administration with an intraperitoneal injection of ketamine/xylazine 10:1, 5 min before blood collection. Whole blood (approximately 1.5 mL) samples were taken by cardiac puncture at 0.3, 0.6, 1, 1.2, 2, 2.5, 3, 4, 5, 7, and 24 h post-treatment. Blood was collected into heparinized vials and centrifuged at 9000 rpm for 15 min to obtain plasma, which was transferred to a plastic tube and stored at −20 °C until HPLC analysis.

#### 2.9.1. HPLC Analysis

The MBZ plasma concentration of each mouse was quantified by HPLC analysis using a UVDAD detector (Agilent 1260 HPLC DAD). Briefly, each plasma sample (200 μL) was deproteinized by adding methanol (1 mL), frozen for 30 min at −20 °C, and centrifuged at 15,000 rpm at 4 °C for 12 min. Then, the supernatant was evaporated under vacuum, and the dried samples were stored frozen until analyzed. Samples were reconstituted in 100 μL of the mobile phase and filtered through 0.2 μm polyvinylidene fluoride (PVDF) syringe filters (Agilent, Santa Clara, CA, USA). The mobile phase consisted of 0.05 M of monobasic potassium phosphate buffer and methanol (40:60, *v*/*v*). The MBZ plasma concentration of each mouse was quantified by HPLC analysis using a UVDAD detector (Agilent 1260 HPLC DAD). Briefly, each plasma sample (200 μL) was treated to eliminate proteins by adding methanol (1 mL), frozen for 30 min at −20 °C, and centrifuged at 15,000 rpm at 4 °C for 12 min. Then, the supernatant was evaporated under a vacuum, and the dried samples were frozen until analyzed. Samples of 25 μL were injected, and the analytes were eluted (flow 1.2 mL/min) from a column Zorbax Eclipse C-18 (150 × 4.6 mm; 5 μm). The absorbance was measured at a wavelength of 247 nm. The analyte retention time was 3.8 ± 0.2 min. A calibration curve was made with a control concentration of MBZ, at a range of 0.25–4 µg/mL (R^2^ = 0.99). MBZ was determined by comparing retention times with those of a pure reference standard [35].

According to regulatory guidelines, the Lower Limit of Quantification (LLOQ) is defined as the lowest concentration of an analyte in a biological matrix that can be quantitatively determined with acceptable precision and accuracy. To establish the LLOQ for MBZ, several concentrations were tested during the calibration curve development. The LLOQ was determined based on the standard deviation of the response at the lowest concentration point, measured in triplicate, and was evaluated for both precision (% CV) and accuracy criteria. Based on these results, MBZ concentrations measured beyond 7 h post-dose were consistently found to be below the LLOQ, and therefore were not reliably quantifiable. Consequently, only data up to 7 h were considered for pharmacokinetic analysis.

#### 2.9.2. Pharmacokinetic Parameters—Relative Bioavailability

To estimate MBZ bioavailability in the different formulations, the parameters’ maximum plasma concentration (C_max_), time to achieve the maximum plasma concentration (T_max_), and areas under the concentration–time curve from 0 to 7 h (AUC_0–7_) were determined [36]. AUCs were calculated using the trapezoidal rule. Pharmacokinetic parameters were expressed as mean ± SEM. In this section, we report data as mean ± SEM to provide a measure of the central tendency along with the precision of the estimated mean. The standard error of the mean (SEM) is particularly useful for highlighting the reliability of the sample mean as an estimate of the true population mean, which is relevant when evaluating pharmacokinetic (PK) parameters under controlled conditions.

The HPLC method used in this study was adapted from the protocol described by De Ruyck et al. (2003) [35]. To establish the calibration curve, we tested several concentrations. The LLOQ was determined based on the standard deviation of the lowest concentration point, measured in triplicate. For this reason, we only used the measured values up to 7 h.

Pharmacokinetic parameters such as C_max_ and AUC often exhibit inter-individual variability and may not follow a strictly normal distribution, particularly in small or heterogeneous samples. In such cases, mean ± standard deviation (SD) provides a better representation of data dispersion and variability within the sample, which is especially relevant when describing population characteristics. We have therefore chosen to report mean ± SD to reflect the variability among subjects, rather than the mean ± standard error of the mean (SEM), which is more appropriate when the goal is to indicate the precision of the mean estimate, such as in inferential statistics or comparisons between groups.

The NP and Comp systems’ relative bioavailability (AUC_r_, %) was calculated for each group as follows:AUC_r_ = AUC_system_/AUC_MBZ_ × 100
where AUC_MBZ_ is the mean value of the corresponding genotype and sex.

As a retrospective complementary analysis, using the results obtained for each dosage form, which include the in vitro variable dissolution efficiency (DE), and the in vivo pharmacokinetic variables AUC_0–7_ and C_max_, from the parallel pharmacokinetic studies performed under controlled conditions, we explored a simple, preliminary correlation between the two data sets. Its purpose was not to establish a validated predictive relationship, but rather to provide a first quantitative approximation to support the selection of promising formulations for advancing to preclinical stages in future experiments (Figure A6 and Figure A7, Appendix B) [37].

### 2.10. In Vivo Analysis of the Anthelmintic Activity of Pure MBZ and Its Formulations

#### 2.10.1. Infection

Since CBi/L and CBi+ mice show significant body weight differences, as already stated, adult animals of both sexes were infected orally with an equivalent dose of 2 *T. spiralis* L1 larvae per g of the host’s body weight. Each animal was weighed before infection, 24–48 h before treatment, and before sacrifice. During the infection course, the general health of each mouse (overall physical condition and behavior) was monitored three times a week [38].

#### 2.10.2. Assessment of the New Formulations’ Therapeutic Efficacy

The anthelmintic activity of pure MBZ and its new formulations was evaluated in the parenteral stage of the parasite cycle. A single daily oral dose of MBZ, NP, or Comp was given to each mouse (15 mg MBZ/kg body weight/day) in the parenteral phase of the infection on days 27, 28, and 29 p-i. Six mice per line and sex were randomly assigned to the non-treated control (C) or treated groups (MBZ, NP, or Comp). Seven days after the last dose, mice were sacrificed to quantify encysted muscle larvae in the tongue. The tongue was excised, weighed, and digested overnight at 37 °C in a pepsin–HCl solution. Digestion was interrupted by adding saline, and the larvae were rinsed several times with saline to remove debris. After low-speed centrifugation (250–300 g, 5 min), the supernatant was carefully removed, and the pellet containing larvae was resuspended in 2 mL of saline. Larvae were counted under an optical microscope at 40× magnification using an acrylic plate. Each formulation’s anthelmintic efficacy was assessed by its effect on the larval muscle load (relative larval load: rLL, number of L1 larvae per g fresh tissue). Also, the effect of the formulations on the encysted L1 larvae viability (dead larvae percentage: proportion of dead muscle larvae/total muscle larvae recovered from each animal) was analyzed using the methylene blue supravital staining technique, as already described [39].

#### 2.10.3. Assessment of Increasing Doses of Pure MBZ on Its Therapeutic Efficacy

Higher concentrations of pure MBZ were administered to determine whether increasing the dose significantly improved the drug’s therapeutic outcome. Briefly, adult susceptible CBi+ mice of both sexes orally infected with 2 *T. spiralis* L1 larvae/g of the host’s body weight were randomly assigned (n = 4 per sex, per treatment) to non-treated control (C) or treated groups receiving a daily dose of 30 or 45 mg MBZ/kg body weight. As in the experiments described in Section 2.10.2, the pure drug was administered in the parenteral stage of infection on days 27, 28, and 29 p-i. Seven days after the last dose, mice were sacrificed, and the number and viability of the tongue’s encysted muscle larvae were determined to evaluate anthelmintic efficacy. The results were compared with those obtained from animals treated with the pure drug or the formulations at a dose of 15 mg MBZ/kg body weight.

### 2.11. Statistical Analysis

Survival curves were calculated using Kaplan and Meier’s product limit method (GraphPad Prism^®^ 9.5; GraphPad Software, Inc., San Diego, CA, USA). The significance of the difference among survival curves was determined with the log-rank test.

The statistical significance of the differences among treatment groups was assessed with a one-way analysis of variance (ANOVA), followed by the Tukey post-test or the nonparametric Kruskal–Wallis’ test and Dunn’s test, as appropriate [40]. Comparisons between sexes within genotype and treatment were analyzed with Student’s *t*-test or the nonparametric Mann–Whitney test. Differences were considered significant if *p* < 0.05.

## 3. Results and Discussion

### 3.1. Particle Size, Solubility, and Dissolution Studies

The physicochemical characteristics of the new formulations were studied, and particle size, solubility, dissolution efficiency (DE), and dissolution rate (DR) were measured (Table 1).

The novel systems, nanoparticles (NPs) and β-cyclodextrin inclusion complexes (Comp), showed decreased particle size and improved solubility and DR compared with the pure drug. In both formulations, MBZ particles’ size decreased by 2–98 times: Comp exhibited particles with a less than 5 µm mean diameter, and NPs were under 500 nm and monodispersed. NPs increased MBZ solubility and DR, but to a much lower extent than the pure drug. Comp presented the best dissolution profile, indicating that β-cyclodextrin citrate should be the excipient of choice to improve the solubility of poorly water-soluble active pharmaceutical ingredients. It should be expected that, as particle size decreases, the aqueous solubility properties of the new orally bioavailable MBZ formulations improve positively, thereby impacting the drug’s systemic absorption [41].

### 3.2. In Vitro Assay

Figure 1 and Table 2 show the effect of NP and Comp on the survival curves and survival parameters of the female parasites cultured for 30 h in medium RPMI 1640 containing the MBZ formulations. The Kaplan–Meier estimates revealed a lower survival time for the new MBZ systems, thus improving MBZ parasiticidal activity (*p* = 0.023). The NP formulation was considerably more efficient than pure MBZ (*p* = 0.009), with a median survival of 30 h and a proportion of live worms of 47.6% at the end of the experiment; the survival estimates of Comp did not differ significantly from that of the pure drug in the period analyzed.

The effect of the formulations on the newborn larvae released by *T. spiralis* females exposed to the antiparasitic solutions was also examined. Neither MBZ nor the formulations affected the mobility of the *T. spiralis* newborn larvae during the studied period, notwithstanding that fewer larvae were observed in the wells containing the NP system.

It is known that increasing the solubility of an active pharmaceutical ingredient can be achieved by “particle size reduction”, which leads to an increase in surface area and a faster dissolution rate [42,43,44,45]. The particle size of the nanoparticles in this formulation was small enough to facilitate the rapid passage of the drug through the cuticle of *T. spiralis* females, thus disrupting the reproductive capacity of the female worms and producing some degree of infertility during the in vitro assay.

The NP formulation’s significant improvement in efficacy suggests that it could be a promising alternative to traditional MBZ treatments.

### 3.3. Pharmacokinetic Analysis

The NP and Comp systems’ bioavailability was evaluated by measuring MBZ plasma concentration levels and comparing them to pure MBZ. Figure 2 shows the plasma levels of MBZ as a function of time after a single oral dose of pure MBZ, NP, or Comp given to CBi/L and CBi+ animals of both sexes. Table 3 and Table 4 summarize the pharmacokinetic parameters, maximum plasma concentration (C_max_), time to reach the maximum concentration (T_max_), and area under the plasma-drug-concentration-versus-time curve (AUC), derived from the bioavailability curves for CBi/L and CBi+ mice, respectively. 

Two of the three bioavailability parameters examined in this analysis, C_max_ and AUC, showed a genotype effect in the MBZ group: CBi/L mice had significantly higher C_max_ (♂ *p* = 0.044; ♀ *p* = 0.007) and AUC (♂ *p* = 0.085; ♀ *p* = 0.014) values than CBi+ animals. The variation in the pharmacokinetic response observed in this group is only attributable to the host’s genotype [46,47] since the parameters did not show a sex effect.

Formulating drugs with poor water solubility as nanoparticles or cyclodextrin complexes has proven to be an effective strategy for increasing the drug’s plasma levels.

This enhancement directly addresses the limitations of poor bioavailability, allowing for improved systemic drug exposure and therapeutic outcomes [38,48]. In this animal model, the effect of NP or Comp on MBZ bioavailability depended on both the host genotype and sex. In CBi/L mice (Table 3), NP led to a higher plasma concentration and a significantly increased C_max_ compared to animals receiving pure MBZ (♂ *p* = 0.04; ♀ *p* = 0.01); mice given Comp showed a nonsignificant increase in the parameter (*p* > 0.05). Conversely, in CBi+ mice (Table 4), both NP and Comp led to a higher C_max_ than those treated with pure MBZ (♂ *p* = 0.0002; ♀ *p* < 0.0001). Notably, only CBi+ males showed differences between formulations: NP improved C_max_ compared with those receiving Comp (♂ *p* = 0.0029). This difference between formulations was not observed in CBi+ females nor CBi/L animals (*p* > 0.5), thus implying a genotype x sex interaction in the MBZ formulations’ bioavailability [49]. T_max_ values also showed genotype and sex effects in the groups receiving the formulations. Regardless of genotype, females given NP or Comp doubled the T_max_ value obtained in the MBZ group; CBi/L males administered the new systems did not modify T_max_, while CBi+ males increased it 3- to 4-fold (Table 4). These results also reveal a genotype x sex interaction for Tmax. In summary, the rate of absorption of MBZ estimated by both T_max_ (an indirect and approximate measure of the duration of the absorption process) and C_max_ (partly determined by the balance between rates, rate of entry, and rate of exit) was affected by the sex and genotype of the host in this animal model.

Differences attributable to genotype were also observed in bioavailability after oral administration, estimated by computing the AUC. AUC increased in all CBi/L groups (Table 3), as reflected in AUCr; however, only CBi/L males given NP had a significantly higher AUC_0–7_ than those receiving pure MBZ (*p* = 0.004). CBi+ males and females treated with NP or Comp (Table 4) significantly increased the AUC_0–7_ compared to pure MBZ (♂ *p* = 0.015; ♀ *p* = 0.006), but there were no significant differences between the formulations (*p* > 0.05). Although AUC_0–7_ did not show a significant sex effect (*p* > 0.05) in any formulation, in both genotypes, the relative bioavailability (AUCr) showed differences between sexes. Males given NP or Comp had a higher average increase in AUCr_0–7_ than females of the same genotype and treatment group.

Biological sex is a significant source of variation in the responses to pharmacological treatments due to its effect on genes involved in the metabolism and transport of the active pharmaceutical ingredient and excipients [50]; however, the magnitude of sex differences depends on the substrates and metabolic pathways involved [51]. Additionally, recent reviews have emphasized the role of sex as a biological variable significantly influencing the efficacy and toxicity of therapeutic nanomaterials. There is compelling new evidence that sex differences can alter the efficacy of nanoparticles at the cellular level. Among other factors, the level and pathway of nanoparticle uptake and intracellular trafficking in certain human cells are heavily influenced by sex; furthermore, the composition of the biomolecular/protein relationship is affected by sex-specific paracrine factors [52]. Anatomical and physiological differences between sexes impact nano drugs’ four main processes: absorption, distribution, metabolism, and excretion [53]. There are some differences in the functioning of the gastrointestinal system between males and females. Female sex hormones, gastric hormone receptors, and nitric oxide influence gastric motility, likely resulting in decreased contraction frequency and slower gastric emptying, as observed in rats. Sex-dependent differences in body composition can also affect the distribution of pharmacological or non-medicinal formulations: males and females have a distinct total body water distribution and differences in muscle and fat distribution.

The observed differences could also involve cytochrome P450 enzymes. *CYP* genes are highly polymorphic and represent a significant source of variability in pharmacokinetics and drug response. Additionally, several *CYP* genes have been documented to have sex-dependent expression or activity in rodents and humans [54,55,56]. CYP3A4 is a critical hepatic cytochrome that metabolizes up to 50% of all drugs and is the most abundant CYP isoenzyme in the human liver. Most clinical studies indicate that women metabolize drugs faster than men [57], as was observed in our animal model; though not significant, females tended to achieve a lower AUC than males. Recognizing the sources and understanding the factors contributing to pharmacokinetic and pharmacodynamic variability within and between individuals remains a significant challenge, especially for drugs with a narrow therapeutic index [58,59].

### 3.4. In Vivo Anthelmintic Activity of MBZ Formulations

The in vivo therapeutic efficacy of the MBZ formulations and the comparison of their effectiveness with that of the pure drug were assessed in the chronic stage of the infection with *Trichinella spiralis* in lines CBi/L and CBi+ of the CBi-IGE animal model of trichinellosis [34]. During this study, animals neither significantly changed their body weight nor showed signs of impaired health status. Also, signs or symptoms of trichinellosis or adverse effects resulting from the treatments were not observed.

The level of infection in each animal, measured through the relative muscle parasite burden (rLL), showed significant differences between lines. As expected, CBi/L mice of the control group showed significantly lower rLL than CBi+ controls (*p* < 0.001). This result is directly related to genotypic differences between these lines: CBi/L behaves as resistant against infection with *T. spiralis* and has a very low parasite load compared to the highly susceptible CBi+ genotype. Contrary to what other authors have observed, treatment with pure MBZ on days 27, 28, and 29 p-i did not reduce the number of muscle larvae recovered compared to the control animals. The different responses could result from using dissimilar treatment protocols, mainly higher doses of MBZ, either administered twice a day or for more extended periods [60,61,62].

As stated by Wesołowska [63], the sex of a host affects the intensity, prevalence, and severity of helminth infection. One sex may be more susceptible than the other, with the prevalence and intensity of helminth infections being generally higher among male hosts than female hosts. Nevertheless, exceptions may exist. Table 5 shows that rLL on day 37 p-i was significantly lower in animals treated with NP or Comp compared to control or pure MBZ groups, regardless of genotype and sex. However, CBi/L females and males did not respond equally to treatment; females showed a higher muscle larval load than males in the three treatment groups (*p* < 0.05). This sex effect was not observed in treated CBi+ mice or the control groups.

The improvement in therapeutic efficacy achieved with the MBZ formulations was also observed when determining the viability of the recovered muscle larvae. CBi/L and CBi+ control animals showed significant differences in the proportion of dead L1 larvae; however, the treatment groups did not exhibit this genotype effect. NP- and Comp-treated males and females from both lines increased the proportion of dead larvae, as estimated using the methylene blue test. This increase was significant in CBi+ males and females and CBi/L females when compared to their controls or MBZ-treated groups (CBi+: ♂ *p* = 0.0043; ♀ *p* = 0.0005; CBi/L: ♀ *p* = 0.0122). CBi/L males given the formulations also showed an increase in the proportion of dead larvae, but the difference was not statistically significant. This lack of significance would be, in part, a consequence of a reduction in the sample size since L1 larvae were not recovered in three out of the six male mice studied in each treated group.

CBi+ males and females treated with 30 or 45 mg of MBZ /kg showed a 40% decrease in muscle rLL compared with the control mice or those treated with 15 mg of MBZ/kg (Figure 3).

However, despite increasing the dose of pure MBZ, animals treated with NP or Comp still showed a significantly lower muscle load (♂, *p* < 0.0001; ♀, *p* = 0.0036) and a higher dead larvae percentage compared with the pure MBZ (♂, *p* = 0.0005; ♀, *p* < 0.0001). The efficacy of increased MBZ dosage in treating *T. spiralis* infection was low compared to what other investigators have reported. The differences observed across experiments may reflect variations in treatment regimens, parasite strains, or experimental models.

Figure 4 shows dead larvae recovered from mice of either sex or genotype treated with NP or Comp, which exhibited a highly altered internal structure. Furthermore, they were [60,64,65,66] stained in a light blue [67], almost transparent color and localized on the solution’s surface (Figure 4B). In contrast, dead larvae obtained from MBZ-treated animals showed a dark blue stain, maintained the typical internal structure, and were found at the bottom of the plate (Figure 4A).

During the parenteral phase, *T. spiralis* larvae penetrate muscle cells, modifying them and creating a protective niche known as a nurse cell or larval cyst. A new capillary network, originating from pre-existing blood vessels and required for the larvae’s nutrition and waste elimination, forms on the surface of the nurse cell. The larva/nurse cell complex is characterized by promoting chronic inflammation at the site of infection, sustained by the “invasion” of the infected muscle tissue by the host’s immune cells that attempt to destroy it [68]. Histological studies by other authors concluded that MBZ acts primarily on the tissue surrounding the larvae, causing larval metabolites and antigens to enter the host’s circulation, eliciting more intense immune and inflammatory responses. MBZ induces significant degenerative changes in the nurse cells, allowing inflammatory cells to infiltrate and ultimately destroy the parasite. The anthelmintic action of MBZ disrupts the cellular components of the cell matrix, turning each larva into an exposed antigenic focus that triggers cell-mediated immunity. Notably, 80 to 100% of the larvae recovered from CBi/L and CBi+ males and females treated with NP and Comp formulations were dead and showed destroyed internal structures.

In contrast, the few dead larvae recovered from animals treated with MBZ retained their internal conformation. If the anthelmintic action of MBZ in the muscle phase is the result of combined pharmacological and immunological effects, it could be hypothesized that a larger number of dead larvae would stimulate a stronger immune response, leading to the destruction of the parasite’s internal structure, potentially resulting in the total disintegration of the larva. Such a situation could explain the results obtained in male CBi/L mice, in which no muscle larvae were recovered, and the differences in the structure of the dead larvae observed in each treatment group (Figure 4).

While MBZ has demonstrated a considerable effect on all stages of the *T. spiralis* life cycle in mice treated with high doses, its use would be limited in clinical practice since it has been found to be effective only against newborn larvae present in blood vessels and lymphatics, but not against larvae encapsulated in muscle cells [69]. Treatment of human trichinellosis is a complex matter since antiparasitic treatment, the control of inflammation, analgesic and symptomatic medication, and rehabilitation in the case of chronic trichinellosis should be considered [70]. Prolonged oral high-dose MBZ therapy may provide an effective anthelmintic response, but side effects such as Herxheimer-like reactions may occur after the initiation of the specific treatment. These reactions are usually observed following the massive discharge of antigenic molecules and immunologic stimulation and can also be noticed in heavy infections. It should be stressed that MBZ is not recommended for treating human trichinellosis in the muscle phase, as exacerbating the immune response generated by the presence of the encysted larvae could provoke, as noted above, a reaction that is similar to a dangerous anaphylactic shock [71].

## 4. Conclusions

The results showed that both formulations improved the therapeutic efficacy of MBZ in genetically different hosts, reducing the parasitic load by 80–90%, compared with the corresponding control groups. In the animal model employed, the host’s genotype contributed noticeably to the expression of sex differences in pharmacokinetic and therapeutic responses. The bioavailability analysis of pure MBZ revealed that host genotype and sex significantly influenced the pharmacokinetic parameters of the drug. While these effects were observed to a lesser degree in animals who were administered the formulations, they still played a role in the absorption profile. The significant increase in the anthelmintic efficacy of MBZ against already encysted *T. spiralis* parasites would be primarily attributed to the improved absorption provided by these novel formulations, which overcome the limitations of the drug’s poor solubility and low bioavailability, and result in higher plasma concentrations of the active drug, even at low doses. These findings suggest that the MBZ formulations developed in our lab are suitable for treating *T. spiralis* infection, highlighting a potential improvement in the pharmacological treatment of trichinellosis.

## Figures and Tables

**Figure 1 pharmaceutics-17-01069-f001:**
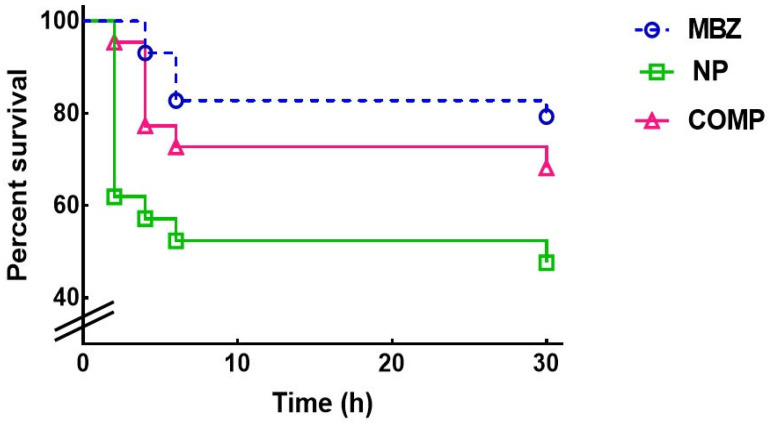
Analysis of the in vitro parasiticidal effect of the MBZ pure drug and the new formulations on *T. spiralis* adult worms. Survival curves of *T. spiralis* females cultured with the different MBZ antiparasitic solutions were generated using the Kaplan–Meier product limit method. The significance of the differences between curves was estimated with the log-rank test.

**Figure 2 pharmaceutics-17-01069-f002:**
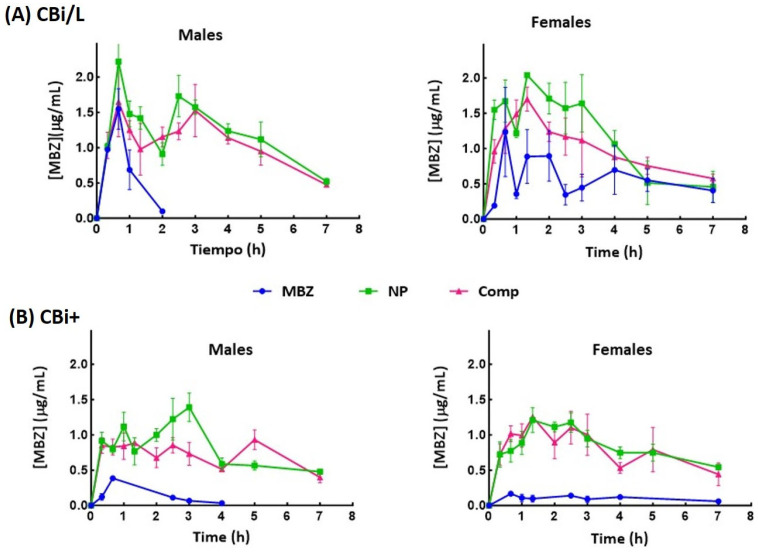
Plasma MBZ concentration–time profiles after oral administration of a single dose of pure MBZ, NP, or Comp (dose, 15 mg MBZ/kg body weight) to male and female mice of the lines CBi/L (**A**) and CBi+ (**B**). Each time point represents the mean ± SEM of 3 mice.

**Figure 3 pharmaceutics-17-01069-f003:**
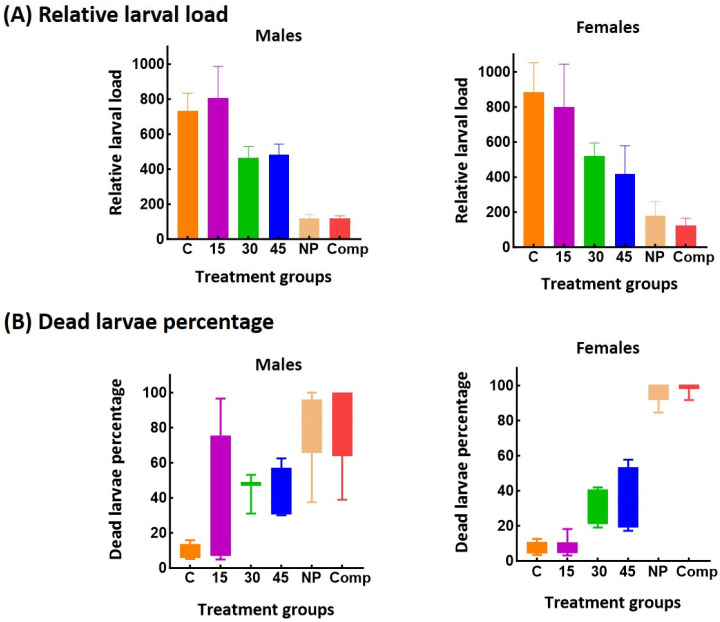
Comparative study of the therapeutic efficacy of increasing mebendazole dose vs. a low dose of the new formulations in *T. spiralis*-infected CBi+ mice. The drug was administered in the chronic stage of the infection. (**A**) Relative larval load: total muscle-encysted larvae per g of fresh tissue. Larval load was measured in the tongue, a preferred site of encystment in mice. (**B**) Dead larvae percentage: proportion of dead muscle larvae/total muscle larvae recovered from each animal. The significance of the differences among groups was evaluated by a one-way ANOVA, using Bonferroni’s post-test for comparisons between groups (Relative larval load) or by the nonparametric Kruskal–Wallis’ test and Dunn’s test for between-group comparison (Dead larvae percentage).

**Figure 4 pharmaceutics-17-01069-f004:**
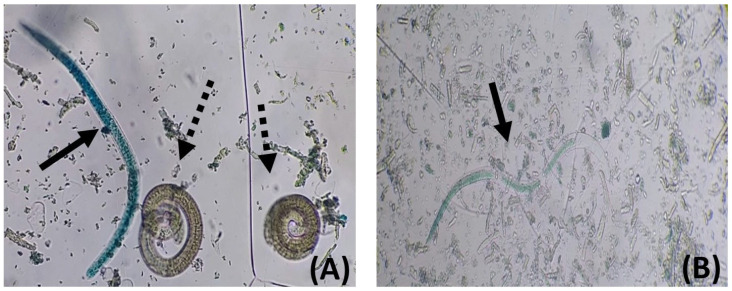
Representative micrographs of *Trichinella spiralis* L1 larvae recovered after artificial digestion of the tongue of mice in the chronic stage of the infection, stained with the methylene blue supravital technique (magnification 40×). (**A**) The solid arrow indicates a dead L1 larva obtained from a mouse treated with MBZ showing the characteristic “comma” shape and blue color; dashed arrows point to live, unstained L1 larvae. (**B**) Dead, non-viable L1 larva from an NP-treated mouse exhibiting an altered internal morphology and light blue staining (solid arrow).

**Table 1 pharmaceutics-17-01069-t001:** Particle size, solubility, dissolution efficiency, and in vitro IC_50_ values after 72 h of exposure to the different MBZ formulations in medium with a pH of 7.2.

Formulation	Particle Size # (µm)	Solubility in HCl 0.1 N (mg/mL)	Solubility Increase (Fold)	Dissolution Efficiency (%)	Dissolution Efficiency Increase (Fold)	IC50 (μM)
MBZ	10.4 ± 6.9	0.025 ± 0.003	-	20.6 ± 0.1	-	>200
MBZ-PVA	0.106 ± 0.004 (PI 0.083 ± 0.022) Zeta potential 15.63/−23.6 mV	0.53 ± 0.04	21	74.2 ± 0.5	3.6	>200
MBZ-CD	4.2 ± 2.1	2.015 ± 0.009	81	87.4 ± 0.3	4.2	>200

MBZ-PVA: nanoparticulate formulation; MBZ-CD: cyclodextrin inclusion complex; **#** mean diameter; PI: polydispersity index; PI < 0.5 monodisperse; PI > 0.5 polydisperse.

**Table 2 pharmaceutics-17-01069-t002:** Effect of the MBZ formulations on the survival parameters of *T. spiralis* females after 30 h of incubation in RPMI 1640 medium containing the MBZ systems.

Formulations	Median Survival (hours)	Survival Proportion After 30 h (%)
MBZ ^a^	Undefined	79.3
NP ^b^	30	47.6
Comp (inclusion complex) ^a^	Undefined	68.2

Data in the table are derived from the Kaplan–Meier survival curves generated with GraphPad Prism version 9.5. The significance of the differences among formulations was calculated with the log-rank test. Groups not sharing the same superscript differ significantly (*p* < 0.025).

**Table 3 pharmaceutics-17-01069-t003:** Pharmacokinetic parameters for MBZ obtained after oral administration of MBZ, NP, or Comp to male and female CBi/L mice *.

Parameter	Sex	Formulations		
		MBZ	NP	Comp
C_max_ (µg/mL) #	♂	1.3 ± 0.31 ^a^	2.3 ± 0.28 ^b^	1.7 ± 0.42 ^a, b^
	♀	1.3 ± 0.26 ^a^	2.2 ± 0.08 ^b^	1.7 ± 0.20 ^a, b^
T_max_ (h)	♂	0.66	0.66	0.66
	♀	0.66	1.33	1.33
AUC_0–7 h_ (µg h/mL) #	♂	2.7 ± 0.82 ^a^	8.2 ± 0.34 ^b^	5.1 ± 1.00 ^a, b^
	♀	3.9 ± 0.63 ^a^	7.4 ± 1.16 ^a^	6.4 ± 1.24 ^a^
AUCr_0–7 h_ (%)	♂	---	204	88
	♀	---	90	64

* Mice were given a single oral dose of 15 mg MBZ/kg body weight. MBZ: pure mebendazole; NP: MBZ nanoparticles; Comp: MBZ inclusion complex with β-cyclodextrin citrate; # Mean ± SEM: differences among the treatment groups within sex were estimated using a parametric ANOVA followed by Tukey’s multiple comparisons test. In each row, groups that do not share the same superscript are significantly different (*p* < 0.025).

**Table 4 pharmaceutics-17-01069-t004:** Pharmacokinetic parameters for MBZ obtained after oral administration of MBZ, NP, or Comp to male and female CBi+ mice *.

Parameter	Sex	Formulations		
		MBZ	NP	Comp
C_max_ (µg/mL) #	♂	0.3 ± 0.11 ^a^	1.5 ± 0.08 ^b^	1.1 ± 0.04 ^c^
	♀	0.2 ± 0.02 ^a^	1.3 ± 0.09 ^b^	1.4 ± 0.08 ^b^
T_max_ (h)	♂	0.66	3	5
	♀	0.66	1.33	1.33
AUC_0–7 h_ (µg.h/mL) #	♂	0.4 ± 0.20 ^a^	5.4 ± 0.18 ^b^	4.7 ± 0.50 ^b^
	♀	0.6 ± 0.08 ^a^	5.7 ± 0.35 ^b^	5.3 ± 0.63 ^b^
AUCr_0–7 h_ (%)	♂	---	1250	1175
	♀	---	850	783

* Mice were given a single oral dose of 15 mg MBZ/kg body weight. MBZ: pure mebendazole; NP: MBZ nanoparticles; Comp: MBZ inclusion complex with β-cyclodextrin citrate; # Mean ± SEM: differences among the treatment groups within sex were estimated using a parametric ANOVA followed by Tukey’s multiple comparisons test. In each row, groups that do not share the same superscript are significantly different (*p* < 0.025).

**Table 5 pharmaceutics-17-01069-t005:** Effect of MBZ or its formulations given in the chronic stage of the infection on the number of muscle-encysted *T. spiralis* L1 larvae.

		CBi/L		CBi+	
		Male	Female	Male	Female
	Variable	Relative larval load §			
Treatment					
Control		95 ± 27.6 ^a^	101 ± 29.5 ^a^	554 ± 104.2 ^a^	952 ± 266.7 ^a^
MBZ		102 ± 40.2 ^a^	222 ± 29.2 ^b^	807 ± 179.5 ^a^	800 ± 243.7 ^a^
NP		10 ± 4.9 ^b^	42 ± 11.8 ^c^	121 ± 20.8 ^b^	180 ± 79.7 ^b^
Comp		15 ± 5.5 ^b^	32 ± 5.4 ^c^	118 ± 16.4 ^b^	122 ± 43.1 ^b^
	Variable	Dead larvae percentage #			
Treatment					
Control		0 (0–3) ^a^	0 (0–4) ^a^	9 (6–12) ^a^	7 (3–10) ^a^
MBZ		27 (7–75) ^b^	15 (6–25) ^b^	11 (7–75) ^a^	5 (5–11) ^a^
NP		100 (0–100) ^b^	82 (38–100) ^c^	82 (66–96) ^b^	100 (92–100) ^b^
Comp		67 (0–100) ^b^	88 (50–100) ^c^	84 (64–100) ^b^	100 (98–100) ^b^

Mice were treated on days 27, 28, and 29 p-i and were sacrificed on day 37 p-i. Relative larval load: total number of muscle-encysted larvae per g of fresh tissue. Larval load was measured in the tongue, a preferred site of encystment in mice. § Mean ± SEM, Dead larvae percentage: proportion of dead muscle larvae/total muscle larvae recovered from each animal. # Median (range): Differences among groups were evaluated by a one-way ANOVA, using Bonferroni’s post-test for comparisons between groups (Relative larval load) or by the nonparametric Kruskal–Wallis’ test and Dunn’s test for between-group comparison (Dead larvae percentage). For each variable, differences between treatment groups not sharing the same letter as superscript are significant at the 0.001 level.

## Data Availability

Data will be made available on request.

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
