# Peer review of "Significant Improvement in Bioavailability and Therapeutic Efficacy of Mebendazole Oral Nano-Systems Assessed in a Murine Model with Extreme Phenotypes of Susceptibility to Trichinella spiralis"

_pharmaceutics, 2025, doi:10.3390/pharmaceutics17081069_

Round 1
Reviewer 1 Report (New Reviewer)
Comments and Suggestions for Authors
This research focuses on the role of two MBZ formulations: NP and COMP, in improving drug bioavailability and antiparasitic effects. The article has a solid amount of data, a rich control group, and results that support the conclusions. However there are many errors in the article, which I hope will be taken seriously and corrected.
1. Some of the text in the main text is bolded (not the title), I don't understand the significance of this?
2. Revise paragraph formatting for line 136.
3. Amend punctuation in line 155.
4. MBZ (0.56 mol); C-β-CD (0.56 mol) in rows 183-184, please add weight units, consistent with above.
5. For line 228, please add the unit of weight.
6. “(n = 93-99 per line and sex)” What does it mean? Please describe it clearly.
7. Is the Zeta potencial 15,63/-23,6 mV in Table 1 correct? Also is Zeta potencial appropriate in this column?
8. Should Figure 1 be supplemented with data on blank vectors for NP and COMP?
9. “Figure 4 shows dead larvae recovered from mice of ....” Should it be Figure 3?
10. “Table 4 shows that rLL on day 37 p-i was significantly lower....” Should it be Table 5?
11. Is there a partial overlap between Figure 5 and Table 4?
Author Response
For research article
|
Response to Reviewer 1 Comments
|
||
|
1. Summary |
|
|
|
Thank you very much for taking the time to review this manuscript. Please find the detailed responses below and the corresponding revisions/corrections highlighted/in track changes in the re-submitted files.
|
||
|
2. Questions for General Evaluation |
Reviewer’s Evaluation |
Response and Revisions |
|
Does the introduction provide sufficient background and include all relevant references? |
Yes/Can be improved/Must be improved/Not applicable |
|
|
Are all the cited references relevant to the research? |
Yes/Can be improved/Must be improved/Not applicable |
|
|
Is the research design appropriate? |
Yes/Can be improved/Must be improved/Not applicable |
|
|
Are the methods adequately described? |
Yes/Can be improved/Must be improved/Not applicable |
|
|
Are the results clearly presented? |
Yes/Can be improved/Must be improved/Not applicable |
|
|
Are the conclusions supported by the results? |
Yes/Can be improved/Must be improved/Not applicable |
|
|
3. Point-by-point response to Comments and Suggestions for Authors |
||
|
Comments 1
This research focuses on the role of two MBZ formulations: NP and COMP, in improving drug bioavailability and antiparasitic effects. The article has a solid amount of data, a rich control group, and results that support the conclusions. However there are many errors in the article, which I hope will be taken seriously and corrected.
Comments 1: Some of the text in the main text is bolded (not the title), I don't understand the significance of this?
|
||
|
Response 1: Thank you for pointing this out. It was a formatting problem not visible in the Word file. We agree with this comment; therefore, we corrected the bold words. |
||
|
Comments 2: Revise paragraph formatting for line 136 |
||
|
Response 2: Agree. We modified the paragraph. . Comments 3: Amend punctuation in line 155. Response 3: Done.
Comments 4: MBZ (0.56 mmol); C-β-CD (0.56 mmol) in rows 183-184, please add weight units, consistent with above. Response 4: Added. The MW of MBZ is 295.29 and the C-β-CD is 1281.36, thus 0.56 mmol of MBZ is 0.165 g and 0.56 mmol of C-β-CD in 0.717 g. These values were added to the text.
Comments 5: For line 228, please add the unit of weight Response 5: Added.
Comments 6: “(n = 93-99 per line and sex)” What does it mean? Please describe it clearly. Response 6: 93-99 is the total number of mice used by line and sex to develop the bioavailability study. 93-99 is the total number of animals of each line and sex used to perform the bioavailability study of the three dosage forms. Blood samples were obtained at 11 different times after drug administration, and, as explained in the text, 3-4 mice of each genotype and sex were sacrificed at each time point.
Comments 7: |
||
|
Is the Zeta potencial 15,63/-23,6 mV in Table 1 correct? Also is Zeta potencial appropriate in this column? Response 7: The table was properly corrected. Also the zeta potential value is correct.
Comments 8: Should Figure 1 be supplemented with data on blank vectors for NP and COMP? Response 8: We prepared blank nanoparticles, to study the efficacy in mice. Comments 9: “Figure 4 shows dead larvae recovered from mice of ....” Should it be Figure 3?
Comments 10: “Table 4 shows that rLL on day 37 p-i was significantly lower....” Should it be Table 5?
Comments11: Is there a partial overlap between Figure 5 and Table 4? Response 11: There is no Figure 5 in the manuscript. |
||

Reviewer 2 Report (New Reviewer)
Comments and Suggestions for Authors
Comments to the authors
The manuscript presents an important and timely study addressing the well-known limitations of mebendazole’s poor aqueous solubility and low oral bioavailability, which constrain its clinical effectiveness against parasitic infections such as Trichinella spiralis. By developing and evaluating two novel oral delivery systems—polyvinyl alcohol-derived nanoparticles (NP) and β-cyclodextrin citrate inclusion complexes (Comp)—the authors aim to enhance the drug's pharmacokinetic and therapeutic profiles. The multidisciplinary approach, combining formulation development, pharmacokinetic analysis, and efficacy testing in a murine model with distinct phenotypic responses, is a notable strength and adds translational value to the study.
However, several sections would benefit from further revision to improve clarity, consistency, and scientific rigor. My specific comments are outlined below:
ABSTRACT: There is a slight repetition in mentioning “enhanced bioavailability” and “improved MBZ absorption” in two separate places. This could be consolidated for a more concise abstract.
INTRODUCTION: While the section includes valuable information, it is at times too lengthy and could be made more concise. Several paragraphs contain redundant details (e.g., multiple references to the same MBZ solubility issue) that could be streamlined to improve readability. Please consider tightening the narrative to keep the focus on the clinical challenge (poor bioavailability of MBZ) and the proposed solutions (NP and cyclodextrin-based formulations), while minimizing tangential or overly technical details unless directly relevant.
Lines 102-104: The solubility values presented here (“poorly water-soluble (1 μg/mL)”) appear inconsistent with those in lines 95–96. Please verify and harmonize these figures for consistency.
Line 114: “with a hydrophobic cavity. and a hydrophilic surface” → should be: “with a hydrophobic cavity and a hydrophilic surface.”
Line 144: “administered by the oral route[27].” → add space before the citation.
Line 147: Missing period after “[28]”. There is some inconsistent spacing, punctuation, and typographic errors that should be addressed throughout.
Line 338: The authors report AUCâ‚€–₇ as the primary pharmacokinetic parameter. Could the authors clarify why AUCâ‚€–₇ was chosen over the more commonly used AUCâ‚€–∞? AUCâ‚€–∞ provides a more complete measure of total drug exposure and is standard in evaluating bioavailability. If AUCâ‚€–₇ was selected due to sampling constraints or specific pharmacokinetic considerations, this should be explicitly stated and justified in the manuscript.
Author Response
For research article
|
Response to Reviewer 2 Comments
|
||
|
1. Summary |
|
|
|
Thank you very much for taking the time to review this manuscript. Please find the detailed responses below and the corresponding revisions/corrections highlighted/in track changes in the re-submitted files.
|
||
|
2. Questions for General Evaluation |
Reviewer’s Evaluation |
Response and Revisions |
|
Does the introduction provide sufficient background and include all relevant references? |
Yes/Can be improved/Must be improved/Not applicable |
|
|
Are all the cited references relevant to the research? |
Yes/Can be improved/Must be improved/Not applicable |
|
|
Is the research design appropriate? |
Yes/Can be improved/Must be improved/Not applicable |
|
|
Are the methods adequately described? |
Yes/Can be improved/Must be improved/Not applicable |
|
|
Are the results clearly presented? |
Yes/Can be improved/Must be improved/Not applicable |
|
|
Are the conclusions supported by the results? |
Yes/Can be improved/Must be improved/Not applicable |
|
|
3. Point-by-point response to Comments and Suggestions for Authors |
||
|
The manuscript presents an important and timely study addressing the well-known limitations of mebendazole’s poor aqueous solubility and low oral bioavailability, which constrain its clinical effectiveness against parasitic infections such as Trichinella spiralis. By developing and evaluating two novel oral delivery systems—polyvinyl alcohol-derived nanoparticles (NP) and β-cyclodextrin citrate inclusion complexes (Comp)—the authors aim to enhance the drug's pharmacokinetic and therapeutic profiles. The multidisciplinary approach, combining formulation development, pharmacokinetic analysis, and efficacy testing in a murine model with distinct phenotypic responses, is a notable strength and adds translational value to the study. However, several sections would benefit from further revision to improve clarity, consistency, and scientific rigor. My specific comments are outlined below: Comments 1: ABSTRACT: There is a slight repetition in mentioning “enhanced bioavailability” and “improved MBZ absorption” in two separate places. This could be consolidated for a more concise abstract.
|
||
|
Response 1: Thank you for pointing this out. We agree with this comment, therefore, we corrected the text. |
||
|
Comments 2: INTRODUCTION: While the section includes valuable information, it is at times too lengthy and could be made more concise. Several paragraphs contain redundant details (e.g., multiple references to the same MBZ solubility issue) that could be streamlined to improve readability. Please consider tightening the narrative to keep the focus on the clinical challenge (poor bioavailability of MBZ) and the proposed solutions (NP and cyclodextrin-based formulations), while minimizing tangential or overly technical details unless directly relevant.
|
||
|
Response 2: Agree. We modified the Introduction. Comments 3: Lines 102-104: The solubility values presented here (“poorly water-soluble (1 μg/mL)”) appear inconsistent with those in lines 95–96. Please verify and harmonize these figures for consistency.
Response 3: Done. Clarified.
Comments 4: Line 114: “with a hydrophobic cavity. and a hydrophilic surface” → should be: “with a hydrophobic cavity and a hydrophilic surface.”
Response 4: Added. Corrected.
Comments 5: “administered by the oral route[27].” → add space before the citation.
Response 5: Added.
Comments 6: Missing period after “[28]”. There is some inconsistent spacing, punctuation, and typographic errors that should be addressed throughout.
Response 6: Corrected.
Comments 7: Line 338: The authors report AUCâ‚€–₇ as the primary pharmacokinetic parameter. Could the authors clarify why AUCâ‚€–₇ was chosen over the more commonly used AUCâ‚€–∞? AUCâ‚€–∞ provides a more complete measure of total drug exposure and is standard in evaluating bioavailability. If AUCâ‚€–₇ was selected due to sampling constraints or specific pharmacokinetic considerations, this should be explicitly stated and justified in the manuscript. Response 7: The rationale for using AUCâ‚€–₇ is stated in lines 321-324 of the corrected manuscript.
|
||

Reviewer 3 Report (New Reviewer)
Comments and Suggestions for Authors
The article describes the improved bioavailability of mebendazole through nanoparticle and β-cyclodextrin citrate formulations. The study investigates its therapeutic effectiveness in murine models. These novel systems significantly improved the drug's solubility and antiparasitic activity against Trichinella spiralis. Increased absorption was demonstrated by pharmacokinetic studies, suggesting the possibility of better trichinellosis treatment. The findings highlight how formulation innovation can help overcome drug solubility limitations.
Please, consider the following suggestions:
Introduction
- Remove the first paragraph on neglected diseases, as it is too generic. Start directly by focusing onTrichinella spiralis and its Public Health relevance.
- Eliminate repetitions, such as the properties of cyclodextrins, which are repeated in successive paragraphs.
- Condense the introduction into 5 or 6 well-structured paragraphs, covering the presentation of the problem, current treatment limitations, promising solutions, the importance of the extreme phenotype animal model, and the study's objective.
- Avoid technical overload in the introduction.
Results and Discussion
- In Table 1, place the values of ZP and PI below the table in the legend; it is unclear what "ABZ" refers to as it is only mentioned once.
- Clarify the terms MBZ-CD and Comp in Table 1, which both indicate improved dissolution profiles. Confirm if both refer to MBZ incorporated into β-cyclodextrin.
- Remove unnecessary horizontal lines in Table 2.
- Compare the results with the literature: The improved therapeutic efficacy of the NP and Comp formulations of mebendazole is evidenced by the significant reduction in parasite load compared to the pure drug. For example, compare the results obtained by Martinez-Fernandez et al. (1987) for alternative formulations.
Comments on the quality of English Language:
Line 116: “Cyclodextrins … has been studied extensively” should read “have been studied extensively” (plural subject).
Line 296: phrases like “mice would eat it without waste” should be revised for tone and clarity (e.g., “ensuring complete ingestion”).
Line 408: sentences are long, leading to decreased clarity. For example, “This variation in the response observed among experiments could be due to different treatment regimens, parasite strains, and experimental models”
Might be reworded as: “The differences observed across experiments may reflect variations in treatment regimens, parasite strains, or experimental models.”
Author Response
For research article
|
Response to Reviewer 3 Comments
|
||
|
1. Summary |
|
|
|
Thank you very much for taking the time to review this manuscript. Please find the detailed responses below and the corresponding revisions/corrections highlighted/in track changes in the re-submitted files.
|
||
|
2. Questions for General Evaluation |
Reviewer’s Evaluation |
Response and Revisions |
|
Does the introduction provide sufficient background and include all relevant references? |
Yes/Can be improved/Must be improved/Not applicable |
|
|
Are all the cited references relevant to the research? |
Yes/Can be improved/Must be improved/Not applicable |
|
|
Is the research design appropriate? |
Yes/Can be improved/Must be improved/Not applicable |
|
|
Are the methods adequately described? |
Yes/Can be improved/Must be improved/Not applicable |
|
|
Are the results clearly presented? |
Yes/Can be improved/Must be improved/Not applicable |
|
|
Are the conclusions supported by the results? |
Yes/Can be improved/Must be improved/Not applicable |
|
|
3. Point-by-point response to Comments and Suggestions for Authors |
||
|
The article describes the improved bioavailability of mebendazole through nanoparticle and β-cyclodextrin citrate formulations. The study investigates its therapeutic effectiveness in murine models. These novel systems significantly improved the drug's solubility and antiparasitic activity against Trichinella spiralis. Increased absorption was demonstrated by pharmacokinetic studies, suggesting the possibility of better trichinellosis treatment. The findings highlight how formulation innovation can help overcome drug solubility limitations. Please, consider the following suggestions: Introduction Comments 1: Remove the first paragraph on neglected diseases, as it is too generic. Start directly by focusing on Trichinella spiralis and its Public Health relevance.
Response 1: Done. Removed.
Comments 2: Eliminate repetitions, such as the properties of cyclodextrins, which are repeated in successive paragraphs.
Response 2: Agree. We modified the Introduction.
Comments 3: Condense the introduction into 5 or 6 well-structured paragraphs, covering the presentation of the problem, current treatment limitations, promising solutions, the importance of the extreme phenotype animal model, and the study's objective. Response 3: Done. Clarified.
Comments 4: Avoid technical overload in the introduction. Response 4: Corrected
Comments 5: In Table 1, place the values of ZP and PI below the table in the legend. Response 5: Corrected.
Comments 6: Clarify the terms MBZ-CD and Comp in Table 1, which both indicate improved dissolution profiles. Confirm if both refer to MBZ incorporated into β-cyclodextrin.
Response 6: Corrected.
Comments 7: Remove unnecessary horizontal lines in Table 2.
|
||
Response 7:
Comments 8: Compare the results with the literature: The improved therapeutic efficacy of the NP and Comp formulations of mebendazole is evidenced by the significant reduction in parasite load compared to the pure drug. For example, compare the results obtained by Martinez-Fernandez et al. (1987) for alternative formulations.
Response 8:
The results obtained by Martínez-Fernández et al. (1987) focus on the characterization of histopathological changes in muscle-stage cysts of Trichinella spiralis in CD-1 mice following treatment with two antiparasitic drugs: mebendazole and niridazole. It is important to note that they did not use any pharmaceutical formulations.
The researchers used CD-1 mice infected with T. spiralis larvae to evaluate the histopathological effects of both drugs. Due to the genetic diversity of CD-1 mice, they offer a more representative model of how a heterogeneous human immune system might respond, which was one of the objectives of their study.
In contrast, our study compares two different mouse strains—one susceptible and one resistant to infection—as well as the influence of gender.
Comments 9: Compare the results with the literature: The improved therapeutic efficacy of the NP and Comp formulations of mebendazole is evidenced by the significant reduction in parasite load compared to the pure drug. For example, compare the results obtained by Martinez-Fernandez et al. (1987) for alternative formulations.
Response 9:
The results obtained by Martínez-Fernández et al. (1987) focus on the characterization of histopathological changes in muscle-stage cysts of Trichinella spiralis in CD-1 mice following treatment with two antiparasitic drugs: mebendazole and niridazole. It is important to note that they did not use any pharmaceutical formulations.
The researchers used CD-1 mice infected with T. spiralis larvae to evaluate the histopathological effects of both drugs. Due to the genetic diversity of CD-1 mice, they offer a more representative model of how a heterogeneous human immune system might respond, which was one of the objectives of their study.
In contrast, our study compares two different mouse strains—one susceptible and one resistant to infection—as well as the influence of gender.
Comments 10: Line 116:“Cyclodextrins has been studied extensively” should read “have been studied extensively”(plural subject).
Response 10: Corrected
Comments 11: Line 296: phrases like “mice would eat it without waste” should be revised for tone and clarity (e.g., “ensuring complete ingestion”).
Response 11: Corrected
Comments 12: Line 408: sentences are long, leading to decreased clarity. For example, “This variation in the response observed among experiments could be due to different treatment regimens, parasite strains, and experimental models”
Response 12: Corrected
Comments 13: Might be reworded as: “The differences observed across experiments may reflect variations in treatment regimens, parasite strains, or experimental models.”
Response 13: Corrected

Reviewer 4 Report (New Reviewer)
Comments and Suggestions for Authors
In this manuscript, the authors presented results of investigation of effects of new formulations of mebendazol on its bioavailability and pharmacological activity. The manuscript has to be improved in order to be reconsidered for publication and here are my suggestion for the improvement of its quality:
- The authors claim in abstract (line 50) and conclusion (line 501) that these are innovative formulations. Nanoparticles and cyclodextrin complexes are well-known and in use for many years. What exactly is novel in your manuscript regarding these formulations?
- Introduction - please give more details on the mechanism of action of mebendazol.
- Do not bold words and sentences in the introduction. Bold has to be removed.
- Page 4 - two sentences do not have the same conclusion (lines 98-99 and 103-104). Does mebendazol have good or poor absorption?
- Lines 104-105 (introduction): the authors wrote that low systemic availability significantly affects mebendazol's therapeutic efficacy. But isn't that in the focus of your research?
- The sentence (lines 110 and 111) is not clear enough and seems to be wrongly positioned.
- Lines 184 - 187: make this sentence easier to understand.
- Lines 210-211: the reference 38 is not correctly presented;
- Lines 218-220: the reference 39 should be positioned at the end of the sentence. This refers to other parts of the manuscript, as well;
- Page 10 - the fact that only 7 hours after the drug administration was monitored is written twice on this page. Please link the second appearance to the first one.
- page 12, table 1: please add more details (into the table title) regarding presented IC50 values. I suppose there is an error in the dissolution efficiency increase values (74.2 and 87.4 are not 7 and 8-fold increased values)? Please correct it.
- Figure 3B - the solid arrow is missing in the figure. Please add.
- Line 495: 80-90% compared to what? Please add.
- Lines 668-669: please correct the reference.
Author Response
For research article
|
Response to Reviewer 4 Comments
|
||
|
1. Summary |
|
|
|
Thank you very much for taking the time to review this manuscript. Please find the detailed responses below and the corresponding revisions/corrections highlighted/in track changes in the re-submitted files.
|
||
|
2. Questions for General Evaluation |
Reviewer’s Evaluation |
Response and Revisions |
|
Does the introduction provide sufficient background and include all relevant references? |
Yes/Can be improved/Must be improved/Not applicable |
|
|
Are all the cited references relevant to the research? |
Yes/Can be improved/Must be improved/Not applicable |
|
|
Is the research design appropriate? |
Yes/Can be improved/Must be improved/Not applicable |
|
|
Are the methods adequately described? |
Yes/Can be improved/Must be improved/Not applicable |
|
|
Are the results clearly presented? |
Yes/Can be improved/Must be improved/Not applicable |
|
|
Are the conclusions supported by the results? |
Yes/Can be improved/Must be improved/Not applicable |
|
|
3. Point-by-point response to Comments and Suggestions for Authors |
||
|
Comments 1: The authors claim in abstract (line 50) and conclusion (line 501) that these are innovative formulations. Nanoparticles and cyclodextrin complexes are well-known and in use for many years. What exactly is novel in your manuscript regarding these formulations? Response 1: We appreciate the reviewer’s comment and agree that both nanoparticles and cyclodextrin complexes are established drug delivery systems. However, the novelty of our study lies in the following aspects:
Comments 2: Introduction - please give more details on the mechanism of action of mebendazol. Response 2: Agree. We modified the Introduction.
Comments 3: Do not bold words and sentences in the introduction. Bold has to be removed. Response 3: Done. It was a formatting issue not visible in the Word file.
Comments 4: Page 4 - two sentences do not have the same conclusion (lines 98-99 and 103-104). Does mebendazol have good or poor absorption? Response 4: Added.
Comments 5: Lines 104-105 (introduction): the authors wrote that low systemic availability significantly affects mebendazol's therapeutic efficacy. But isn't that in the focus of your research? Response 5: Corrected.
Comments 6: The sentence (lines 110 and 111) is not clear enough and seems to be wrongly positioned. Response 6: Corrected.
Comments 7: Lines 184 - 187: make this sentence easier to understand. |
||
Response 7: The sentence was corrected and the paragraph modified as shown in lines 153-165.
Since the formulations were developed for oral administration, hazardous solvents should be avoided, and only Class 3 solvents, as formic acid, should be used. Class 3 solvents are characterized by a low toxic potential according to the ICH guideline Q3C by the European Medicines Agency. The objective of this guideline is to recommend acceptable amounts for residual solvents in pharmaceuticals to ensure patient safety.
Comments 8: Lines 210-211: the reference 38 is not correctly presented;
Response 8: Corrected.
Comments 9: Lines 218-220: the reference 39 should be positioned at the end of the sentence. This refers to other parts of the manuscript, as well;
Response 9: Corrected.
Comments 10: Page 10 - the fact that only 7 hours after the drug administration was monitored is written twice on this page. Please link the second appearance to the first one.
Response 10: Corrected
Comments 11: page 12, table 1: please add more details (into the table title) regarding presented IC50 values. I suppose there is an error in the dissolution efficiency increase values (74.2 and 87.4 are not 7 and 8-fold increased values)? Please correct it.
Response 11: Corrected
Comments 12: Figure 3B - the solid arrow is missing in the figure. Please add.
Response 12: Corrected, and is the Figure 4B
Comments 13: Line 495: 80-90% compared to what? Please add.
Response 13: Corrected
Comments 14: Lines 668-669: please correct the reference.
Response: Corrected.

Round 2
Reviewer 1 Report (New Reviewer)
Comments and Suggestions for Authors
The author made revisions according to the comments. Also, the author made a point-to-point response to the comments. I have no further comment.
Author Response
The author made revisions according to the comments. Also, the author made a point-to-point response to the comments. I have no further comment.
Reviewer 3 Report (New Reviewer)
Comments and Suggestions for Authors
The improvements made to the manuscript were carried out satisfactorily, addressing the key points raised during the review process and contributing to the overall clarity and quality of the study.
Author Response
The improvements made to the manuscript were carried out satisfactorily, addressing the key points raised during the review process and contributing to the overall clarity and quality of the study.
Reviewer 4 Report (New Reviewer)
Comments and Suggestions for Authors
The authors responded to all my questions and suggestions and made appropriate corrections. However, there is one minor error in the Table 1 (dissolution efficiency increase should be 3.6 and 4.2, not 3.7 and 4.3). Please correct it
Author Response
The authors responded to all my questions and suggestions and made appropriate corrections. However, there is one minor error in the Table 1 (dissolution efficiency increase should be 3.6 and 4.2, not 3.7 and 4.3). Please correct it.
Response: Done.
This manuscript is a resubmission of an earlier submission. The following is a list of the peer review reports and author responses from that submission.
Round 1
Reviewer 1 Report
Comments and Suggestions for Authors
Dear authors,
I have carefully reviewed your manuscript and appreciate the effort put into this work. The topic is relevant and addresses an important area of pharmaceutical research. However, I have a few comments and suggestions that may help improve the clarity, completeness, and scientific rigor of the manuscript.
-
The introduction is a bit brief. Please elaborate further by including relevant details about the additives used in the formulation, particularly β-cyclodextrin. Discuss its role and applications in similar research studies to provide better context.
-
Rationale for solubility and dissolution medium: Why were the solubility and dissolution studies conducted in only one medium, i.e., 0.1 N HCl? Please specify its pH and clarify whether this choice aligns with pharmacopoeial standards.
-
Missing data presentation: Kindly include the solubility and dissolution study curves in the Results section to support the findings presented.
-
Stability study recommendation: I strongly recommend performing a stability study, as it is well known that nanoparticles tend to aggregate over time, leading to an increase in particle size and potential loss of efficacy. This is critical for assessing the formulation's long-term stability.
-
In-vitro–in-vivo correlation (IVIVC): Since both in-vitro and in-vivo studies were conducted, it is advisable to report the IVIVC data to strengthen the reliability and translational value of the results.
Thank you.
Author Response
|
Comments 1: · The introduction is a bit brief. Please elaborate further by including relevant details about the additives used in the formulation, particularly β-cyclodextrin. Discuss its role and applications in similar research studies to provide better context. |
|
Response 1:
The introduction was improved, and we added the reviewer suggestions. Thank you for pointing this out. We agree with this comment. Therefore, we have added some information about cyclodextrins. The text is written in red. It can be found – 3, paragraph 2. |
|
· Comments 2: · Rationale for solubility and dissolution medium: Why were the solubility and dissolution studies conducted in only one medium, i.e., 0.1 N HCl? Please specify its pH and clarify whether this choice aligns with pharmacopoeial standards.
|
|
Response 2: The dissolution studies were performed according to the conditions described in the USP Pharmacopeia, 29th Edition. We chose to align with the conditions of the validated USP studies, as this represents the most accurate and standardized method for evaluating in vitro drug release. Solubility testing was conducted in 0.1 N HCl without the addition of a surfactant, as the aim was not to enhance solubility but to evaluate the intrinsic dissolution behavior of the drug.
Comments 3:
· Missing data presentation: Kindly include the solubility and dissolution study curves in the Results section to support the findings presented. Response 3: We improved table 1 with all the data missing. Please see page 10.
|
|
Comments 4 · Stability study recommendation: I strongly recommend performing a stability study, as it is well known that nanoparticles tend to aggregate over time, leading to an increase in particle size and potential loss of efficacy. This is critical for assessing the formulation's long-term stability. · In-vitro–in-vivo correlation (IVIVC): Since both in-vitro and in-vivo studies were conducted, it is advisable to report the IVIVC data to strengthen the reliability and translational value of the results. Response 4: Thank you for your valuable suggestions. • Stability Study: We fully acknowledge the importance of conducting a stability study, particularly given the tendency of nanoparticles to aggregate over time, which can significantly impact particle size distribution and, consequently, the formulation’s efficacy. A comprehensive stability assessment is currently being planned to evaluate key parameters such as particle size, zeta potential, and drug content over time under different storage conditions. This will provide essential insights into the long-term stability and robustness of the formulation. • In-Vitro–In-Vivo Correlation (IVIVC): We agree that establishing an IVIVC is an important step to enhance the translational relevance of our findings. We are in the process of analyzing the available data to determine the correlation between the in vitro dissolution profile and the in vivo pharmacokinetic performance. The outcomes of this analysis will be included in a future version of the manuscript to support the predictive value of the in vitro model and strengthen the overall conclusions.
|

Reviewer 2 Report
Comments and Suggestions for Authors
Overall the manuscript is well written and presents a novel way of improving the solubility and bioavailability of an anthelmintic drug MBZ. The authors evaluated the efficacy of the particle formulations in vitro and in vivo and compared the results to the pure drug and observed significant improvement. Some major and minor concerns are noted below. Major revisions:
- For nanoparticle synthesis (lines 110-116), was formate assay done to ensure there was no residual formic acid remaining in the particles? Presence of formic acid would be toxic and could affect the downstream data collection. Same concern when Comp was made (lines 126-130).
- Measurement of zeta potential to determine particle surface charge would also be valuable in terms of characterizing the particles.
- Report the polydispersity index (PDI) of nanoparticles and list in Table 1.
- Include SEM images of the synthesized NPs and Comp.
- Line 335 : is there data for the particle size for the pure drug? The comparison is unclear especially since no data for the particle size of pure drug is listed in Table 1. Include SEM image of pure drug to show the difference in particle sizes.
- Lines 342-344 : did the authors explore size range of Comp particles? Or is this simply a hypothesis? Some clarification would be useful.
- Authors might want to comment on how despite the improved solubility of MBZ in Comp compared to pure drug, the in vitro assay indicates that Comp is not that different from the pure drug during the 30 h.
Minor revisions:
- Include a reference for water solubility and low bioavailability (lines 65-67)
- One sentence paragraphs (lines 80-81, lines 374-375, lines 408-409). Is it possible to combine these sentences with the previous paragraph?
- Provide standard deviation data for particle size in Table 1
- Include standard error in the graphs plotted in Figure 2.
Author Response
|
Comment 1: · |
|
· For nanoparticle synthesis (lines 110-116), was formate assay done to ensure there was no residual formic acid remaining in the particles? Presence of formic acid would be toxic and could affect the downstream data collection. Same concern when Comp was made (lines 126-130).
|
|
Response 1: Thank you for your insightful observation. The formic acid is the proper solvent to assure the complete solubilisation of the mebendazole. |
|
In the case of nanoparticle synthesis, a formate assay was not initially performed. However, we recognize the importance of confirming the absence of residual formic acid, as its presence could indeed pose toxicity risks and interfere with downstream analyses. To address this, we are currently planning to conduct a formate-specific assay (e.g., using HPLC or enzymatic methods) to quantitatively determine any residual formic acid in the final nanoparticle formulation. The same consideration applies to the Comp preparation. We will ensure that appropriate analytical testing is carried out to confirm the absence of residual formic acid or related impurities, in order to validate the safety and integrity of the final product. These results will be included in future updates to strengthen the study’s quality and reliability. It is also important to highlight that the USP recommends the use of formic acid to ensure the complete dissolution of mebendazole, especially due to its poor aqueous solubility. Therefore, the use of formic acid during sample preparation was intended to follow pharmacopeial guidance and achieve full dissolution for reliable analysis. Nonetheless, all measures will be taken to ensure that any residual formic acid is removed or reduced to acceptable levels in the final formulation. |
|
|
|
· .Comment 2: Thank you for your insightful observation. · Measurement of zeta potential to determine particle surface charge would also be valuable in terms of characterizing the particles. · Response 2: · Done. Please see table 1. · Report the polydispersity index (PDI) of nanoparticles and list in Table 1. Response 2 : Done
Comments 3:
· Include SEM images of the synthesized NPs and Comp. · · Response 3 Thank you for your insightful observation. The images were added. · Please see Appendix A · Comment 4 · Line 335 : Is there data for the particle size for the pure drug? The comparison is unclear especially since no data for the particle size of pure drug is listed in Table 1. Include SEM image of pure drug to show the difference in particle sizes. Response 4: Thank you for your insightful observation. The images were added. · Comments 5: · Lines 342-344 : did the authors explore size range of Comp particles? Or is this simply a hypothesis? Some clarification would be useful. Response 5: Particle Size Analysis Thank you for your insightful observation. The particle size of the active pharmaceutical ingredient (API), nanoparticles and inclusion complexes was determined by scanning electron microscopy (SEM) using a Leitz AMR 1600 T microscope (Amray, Bedford, MA, USA) operating at an acceleration voltage of 20 kV. Prior to imaging, samples were sputter-coated with a thin layer of gold to improve conductivity. SEM images were analyzed using ImageJ software to estimate particle size based on manual measurements from multiple fields of view. In contrast, the particle size of the nanosuspensions was measured using a Nano Particle Analyzer Horiba SZ-100 (Horiba, Germany). Prior to analysis, samples were diluted at a ratio of 1:30 in filtered distilled water to ensure appropriate dispersion and prevent multiple scattering effects. · Commetn 6: · Authors might want to comment on how despite the improved solubility of MBZ in Comp compared to pure drug, the in vitro assay indicates that Comp is not that different from the pure drug during the 30 h. Minor revisions: · Include a reference for water solubility and low bioavailability (lines 65-67) Response 6: Thank you for your insightful observation. The reference was added. · One sentence paragraphs (lines 80-81, lines 374-375, lines 408-409). Is it possible to combine these sentences with the previous paragraph? · · Response 6: Thank you for your insightful observation, it was done. · Comment 7: · Provide standard deviation data for particle size in Table 1 Thank you for your insightful observation. It was done · Include standard error in the graphs plotted in Figure 2. Thank you for your insightful observation, you could see table 3.
|
|
|
|
|
|
|

Reviewer 3 Report
Comments and Suggestions for Authors
Find the attachment

Author Response
|
Comment 1:
|
Mention the prevalence rate of Trichinellosis as per recent reports in the introduction with proper references
Response 1:
Thank you for your insightful observation. It was added to the Introduction, page 2, paragraph 2.
Comment 2.
Briefly include the history and regulatory approval of mebendazole (MBZ) in the introduction and mention the dosage forms currently available on the market.
Response 2
Thank you for your insightful observation. It was added to the Introduction, page 2, paragraph 2.
Comment 3.
Mebendazole (MBZ) is a well-established anthelmintic agent available in various conventional dosage forms. However, please clarify the limitations or shortcomings of these marketed formulations that prompted the development of your proposed strategy. What is the scientific rationale for formulating MBZ into novel nanoparticle-based or complex delivery systems? A clear justification based on pharmacokinetic, biopharmaceutical, or therapeutic advantages is essential.
Response 3
Formulating MBZ into novel nanoparticle-based or complex delivery systems is scientifically justified by several pharmacokinetic, biopharmaceutical, and therapeutic limitations of conventional MBZ formulations. Below is a detailed rationale for this approach:
The main reason is the poor aqueous solubility and oral bioavailability. MBZ is a Class II drug in the Biopharmaceutics Classification System (BCS), characterized by low solubility and high permeability. Its oral bioavailability is extremely low (typically <20%) due to poor dissolution in gastrointestinal fluids. These novel formulations produce an increase in the surface area, improving wettability, and as a consequence improving the dissolution and the absoption.
Comment 4.
What is the rationale for selecting formic acid as the solvent for mebendazole (MBZ) for nanoparticle preparation? Considering that MBZ has high solubility in DMSO why were these solvents not chosen? Additionally, is mebendazole stable under acidic conditions?
Response 4:
|
Thank you for your insightful observation. The formic acid is the proper solvent to assure the complete solubilisation of the mebendazole. |
|
In the case of nanoparticle synthesis, a formate assay was not initially performed. However, we recognize the importance of confirming the absence of residual formic acid, as its presence could indeed pose toxicity risks and interfere with downstream analyses. To address this, we are currently planning to conduct a formate-specific assay (e.g., using HPLC or enzymatic methods) to quantitatively determine any residual formic acid in the final nanoparticle formulation. The same consideration applies to the Comp preparation. We will ensure that appropriate analytical testing is carried out to confirm the absence of residual formic acid or related impurities, in order to validate the safety and integrity of the final product. These results will be included in future updates to strengthen the study’s quality and reliability. It is also important to highlight that the USP recommends the use of formic acid to ensure the complete dissolution of mebendazole, especially due to its poor aqueous solubility. Therefore, the use of formic acid during sample preparation was intended to follow pharmacopeial guidance and achieve full dissolution for reliable analysis. Nonetheless, all measures will be taken to ensure that any residual formic acid is removed or reduced to acceptable levels in the final formulation. |
Comment 5.
In section 2.9.1 (HPLC Analysis), please clarify whether the HPLC method used is a previously reported method or an in-house developed procedure.
Response 5:
The HPLC method used in this study was adapted from the protocol described by De Ruyck et al. (2003).
Comment 6.
In the figure 4A mention statistical difference between different treatment groups
The significance of the differences among groups was evaluated by a one-way ANOVA, using Bonferroni’s post-test for comparisons between groups (Relative larval load) or by the nonparametric Kruskall-Wallis’ test and Dunn’s test for between-groups comparison (Dead larvae percentage).
Comment 7.
In section 2.9 (Pharmacokinetic Analysis), please clarify why a marketed formulation was not selected as the standard comparator. Additionally, could the authors explain the absence of a disease control group in the study design?
Response 7:
Thank you for your insightful observation. During the experimental design phase, the decision was made to compare the novel formulations with the pure drug substance. Therefore, both the newly developed delivery systems and the raw material (unformulated mebendazole) were administered for comparative evaluation.
The disease control group is C in Figure 4 and Table 5.
Comment 8.
In the results section, the dissolution efficiency of MBZ nanoparticles (MBZ-NP) was reported to be lower compared to the MBZ-cyclodextrin complex (MBZ-Comp). However, both in vitro and in vivo studies showed higher activity for MBZ-NP. Could the authors provide a justification for this apparent discrepancy?
Response 8:
Thank you for pointing this out.
This enhancement directly addresses the limitations of poor bioavailability, allowing for improved systemic drug exposure and therapeutic outcomes. In this animal model, the effect of NP or Comp on MBZ bioavailability and therapeutic efficacy depends on both the host genotype and sex
Comment 9.
Since the author has developed two different formulations of mebendazole (MBZ)—a conventional cyclodextrin complex and nanoparticles—please specify which formulation demonstrated superior results overall. This comparison should be clearly highlighted in both the discussion and conclusion sections.
Response 9:
Thank you for your insightful observation. In the conclusions section, we clarify that after carefully analysed the results in this animal model, the effect of NP or Comp on MBZ bioavailability and therapeutic efficacy depends on both the host genotype and sex.
It is dificult to select a formulation.

Round 2
Reviewer 2 Report
Comments and Suggestions for Authors
All revisions adequately addressed.
Author Response
All revisions adequately addressed.